# Cardiomyocyte Differentiation from Mouse Embryonic Stem Cells by WNT Switch Method

**DOI:** 10.3390/cells13020132

**Published:** 2024-01-11

**Authors:** Isaiah K. Mensah, Martin L. Emerson, Hern J. Tan, Humaira Gowher

**Affiliations:** Department of Biochemistry, Purdue University, West Lafayette, IN 47907, USA; imensah@purdue.edu (I.K.M.); tan361@purdue.edu (H.J.T.)

**Keywords:** cardiomyocytes, mesoderm, mouse embryonic stem cells, heart, in vitro differentiation, WNT signaling

## Abstract

The differentiation of ESCs into cardiomyocytes in vitro is an excellent and reliable model system for studying normal cardiomyocyte development in mammals, modeling cardiac diseases, and for use in drug screening. Mouse ESC differentiation still provides relevant biological information about cardiac development. However, the current methods for efficiently differentiating ESCs into cardiomyocytes are limiting. Here, we describe the “WNT Switch” method to efficiently commit mouse ESCs into cardiomyocytes using the small molecule WNT signaling modulators CHIR99021 and XAV939 in vitro. This method significantly improves the yield of beating cardiomyocytes, reduces number of treatments, and is less laborious.

## 1. Introduction

Embryonic stem cells (ESCs) are derived from the inner cell mass (ICM) of the early blastocyst during mammalian development. These ESCs can self-renew almost indefinitely and retain the ability to generate many cell types of the adult mammalian organism under the right conditions in vitro [1,2]. Hence, ESCs are a powerful model for studying mammalian development and disease.

Cardiomyocyte insufficiency, resulting from the inability of cardiomyocytes to regenerate, underlies most heart failures [3,4,5]. Therefore, ex vivo cardiomyocyte generation will significantly contribute to the discovery of novel factors governing the differentiation of cardiomyocytes in mammalian development. Previous methods for differentiating mouse ESCs (mESCs) into cardiomyocytes use growth factors, which are expensive and have short half-lives [6,7]. Here, we describe a relatively inexpensive method to efficiently differentiate mESCs into cardiomyocytes based on temporal changes in WNT signaling observed during cardiomyocyte development.

Seminal studies have highlighted the critical role of canonical WNT/β-catenin signaling in cardiac development [8,9,10,11,12]. The timing for activation and inhibition of WNT signaling during development is crucial for cardiomyocyte differentiation. This includes the early activation of WNT signaling, promoting the differentiation of ESCs into the mesoderm [13,14,15]. The subsequent inhibition of WNT signaling following mesoderm formation commits mesodermal cells into cardiomyocytes [16,17,18,19]. In vitro, small molecule inhibitors can efficiently activate and inhibit WNT signaling [20,21]. The WNT signaling activator, CHIR99021, inhibits GSK3β (glycogen synthase kinase 3 beta) activity, allowing β-catenin to translocate into the nucleus to turn on WNT signaling target genes [22,23]. Similarly, XAV939 is a small molecule that inhibits the WNT signaling pathway by inhibiting tankyrase activity, thereby increasing the axin-GSK3β destructive complex to promote phosphorylation and degradation of β-catenin [21,24]. These small molecule inhibitors are relatively inexpensive, stable, and can access cells embedded deeper in tissues, facilitating a uniform and consistent activation of WNT signaling.

Compared to human ESCs (hESCs) and induced pluripotent stem cells (hiPSCs), mESC differentiation is more frequently used as a mammalian developmental model system for discovering novel mechanisms and factors controlling differentiation and disease. The success of the model system is based on the significant similarities in the early developmental pathways and mechanisms in mammals. Additionally, obtaining and maintaining mESCs are simpler and more affordable than hESCs and hiPSCs and pose fewer administrative tasks [25]. For cardiomyocyte generation, while other methods have utilized small molecules to activate and inhibit WNT signaling genes in hiPSC differentiation [26,27,28,29], these conditions have not been used for mESCs. A recent method used only XAV939 from day 3 to 5 post-differentiation [30], thereby controlling only the inhibition of WNT signaling. In this study, we describe the “WNT Switch” method that uses small molecule inhibitors to temporally control both the activation and inhibition of WNT signaling to induce differentiation of mESC into cardiomyocytes. The differentiation process was characterized by measuring changes in gene expression using RT-qPCR and immunofluorescence. The cardiomyocytes generated were monitored for spontaneous contractile activity using bright-field microscopy, and the contractions per minute were measured. We further quantified the efficiency of differentiation in comparison to previously used methods using fluorescence-assisted cell sorting (FACS). The data support our claim that the “WNT switch” method is robust and generates cardiomyocytes with high efficiency.

## 2. Materials and Methods

### 2.1. Reagents

CHIR99021 (APExBIO, Houston, TX, USA, cat.no. A3011);XAV939 (Stemcell Technologies, Vancouver, BC, Canada, cat.no. 72672);Trypsin-EDTA (0.25%, *w*/*v*; Life Technologies, Carlsbad, CA, USA, cat.no. 25200-056);FBS (Fisher Scientific, Waltham, MA, USA, cat.no. MT35015CV);E14 mouse embryonic stem cells (E14 Tg2A.4, cat.no. 015890-UCD-CELL);NEAA (Thermofisher Scientific, Waltham, MA, USA, cat.no. 11140050);Sodium pyruvate (Thermofisher Scientific, cat.no. 11360070);Penicillin-Streptomycin (Thermofisher Scientific, cat.no. 15140122);L-Glutamine (Thermofisher Scientific, cat.no. 25030081);Nonfat dry milk (can be purchased from a local grocery store);Trypan blue solution (Fisher Scientific, cat.no. 15250061);β-mercaptoethanol (Thermofisher Scientific, cat.no. 21985023);IMDM (Thermofisher Scientific, cat.no. 12440053);DMEM (Thermofisher Scientific, cat.no. 11995065);Evagreen master mix (Fisher Scientific, cat.no. NC1787870);cDNA synthesis kit (Meridian Bioscience, Cincinnati, OH, USA, cat.no. BIO-65043);TRIzol^®^ (Thermofisher Scientific, cat.no. 15596026);Chloroform (Thermofisher Scientific, cat.no. 022920.K2);DNase I (Thermofisher Scientific, cat.no. 18068015);DPBS (Thermofisher Scientific, cat.no. 14200075);Triton X-100 (Millipore, Burlington, MA, USA, cat.no. T9284);Ethanol (Millipore, cat.no. 493511);HEPES (Millipore, cat.no. H4034-25G);Milli-Q water;Primers for quantitative RT-PCR (Appendix A);DAPI solution (Thermofisher Scientific, cat.no. 62248);cTnT antibody (Abcam, Cambridge, UK, cat.no. ab8295);Mouse IgG Alexa Fluor^®^ 488 (Santa Cruz, Dallas, TX, USA, cat.no. sc-3890);Gelatin (Sigma, St. Louis, MO, USA, cat.no. G1393-100 mL);RNase AWAY (Fisher Scientific, cat.no. 501978158);Formaldehyde (Millipore Sigma, Burlington, MA, USA, cat.no. 252549-100 mL);PureLink RNA Mini Kit (ThermoFisher Scientific, cat.no 12183018A).

### 2.2. Equipment

Steriflip filtration system (50 mL; Fisher Scientific, cat.no. SCGP00525);Stericup filtration system (500 mL; Fisher Scientific, cat.no. S2GPT05RE);Zeiss phase-contrast microscope (Spectra Services, Ontario, NY, USA; cat.no. SKU Axiovert35);Fluorescence microscope (Biocompare, South San Francisco, CA, USA cat.no. EVOS^®^FL);Cell culture plates 100 mm (Corning, Glendale, AZ, USA, cat.no. 430167);Petri plates 60 mm (Sigma-Aldrich, St. Louis, MO, USA, cat.no. P5481-500EA);Six-well tissue culture plates (Corning, cat.no. 3516);Tissue culture flask-T25 (Thermofisher Scientific, cat.no. 130189);Hybex bottle flasks (1000 mL, 500 mL; Benchmark Scientific, Sayreville, NJ, USA, cat.no. B3000-100-B, B3000-500-B);Falcon conical tubes (50 mL, 15 mL; Fisher Scientific, cat.no. 14-432-22, 14-959-53A);Cell lifters (Fisher Scientific, cat.no. 08100240);Centrifuge (Eppendorf, Enfield, CT, USA cat.no. 022625080);Biosafety cabinet (Fisher Scientific, cat.no. 13-261-222);Vacuum liquid waste disposal system (Welch, Ilmenau, Germany, cat.no. 2511B-75);Glass Pasteur pipettes (Fisher Scientific, cat.no. 1-678-20D);Serological pipettes (25 mL, 10 mL, 5 mL; Fisher Scientific, cat.no. 13-676-10M, 13-678-11E, 13-676-10H);Humidified tissue culture incubators (37 °C, 5% CO_2_; Eppendorf, cat.no. c170i);Cell counter slides (BIO-RAD, Hercules, CA, USA, cat.no. 1450011);Block heater (Thermofisher Scientific, cat.no. 88870001);Automated cell counter (BIO-RAD, cat.no. 1450102);Water bath incubators (Fisher Scientific, cat.no. 15-462-20Q);Microcentrifuge tubes (1.5 mL; VWR, Batavia, IL, USA, cat.no. 87003-294);Water purification system (Millipore Sigma, cat.no. ZRQSVP3WW);Quantitative-PCR detection system (BIO-RAD, cat.no. 1855201);qPCR 96-well plates (USA Scientific, Ocala, FL, USA, cat.no. 1402-8590).

### 2.3. Reagent Setup

All reagents were prepared in a Biosafety II grade HEPA-filtered biosafety cabinet following UV sterilization for at least 30 min. The benchtop was swabbed with 70% ethanol, and anything going into the biosafety cabinet was also swabbed properly with 70% ethanol. All aseptic procedures were strictly enforced to maintain a contamination-free cell culture. The concentrations in the parenthesis below indicate the final working concentration.

**1000**× **β-mercaptoethanol**: In total, 355 μL of stock β-mercaptoethanol was mixed with 49.65 mL sterile water under sterile conditions. This solution was filtered with the Steriflip filtration system and stored at 4 °C for up to 2 weeks.

**HEPES**: We prepared 50 mL of 1 M HEPES with pH 8 by dissolving 11.92 g of HEPES in 40 mL Milli-Q water. After complete dissolution, the pH 8 of the solution was adjusted using 1 M KOH. The volume of the final solution was brought to 50 mL using Milli-Q water and filter-sterilized with the Steriflip filtration system under sterile conditions. This solution should be stored at 4 °C.

**ES + LIF (Leukemia Inhibitory Factor) Medium**: Under sterile conditions, we prepared 1000 mL of the medium by mixing 804 mL of 1× DMEM, 150 mL of FBS (15%), 10 mL of L-Glutamine (2 mM), 10 mL of NEAA (1×), 10 mL of sodium pyruvate (1 mM), 10 mL of Pen/Strep (1 mM), 1 mL of β-mercaptoethanol (1×), 1 mL of HEPES (1 mM), and 4 mL of LIF (0.4%). The medium was filtered with a 500 mL Stericup filtration system over an autoclaved/sterile 1000 mL bottle. This medium can be stored for up to two months at 4 °C.

**20% FBS Differentiation Medium**: Under sterile conditions, we prepared 500 mL of the medium by mixing 384.5 mL of 1× IMDM, 100 mL of FBS (20%), 5 mL of L-Glutamine (2 mM), 5 mL of NEAA (1×), 5 mL of sodium pyruvate (1 mM), and 500 μL of β-mercaptoethanol (1×). The medium was filtered with a 500 mL Stericup filtration system into an autoclaved/sterile 500 mL bottle. This medium should be protected from light and can be stored for a month at 4 °C.

**Low FBS Differentiation Medium**: Under sterile conditions, we prepared 500 mL of the medium by mixing 488.5 mL of 1× IMDM, 1 mL of FBS (0.2%), 5 mL of L-Glutamine (2 mM), 5 mL of NEAA (1 mM), and 500 μL of β-mercaptoethanol (1×). The medium was filtered with a 500 mL Stericup filtration system into an autoclaved/sterile 500 mL bottle. This medium should be protected from light and can be stored for a month at 4 °C.

**CHIR99021 Medium**: Under sterile conditions, we aliquoted about 30 μL of CHIR99021 into sterile 1.5 mL sterile microcentrifuge tubes and stored them for up to 12 months at −20 °C to prevent freeze–thaw. We added 15 μL of CHIR99021 (3 μM) to 50 mL 20% FBS differentiation medium for differentiation. This medium should be freshly prepared and protected from light. The medium was filtered through the 50 mL Steriflip filtration system for additional sterility.

**XAV939 Medium**: Under sterile conditions, we prepared a 1 mg/mL solution by adding 1 mL of DMSO to 1 mg of XAV939. We used the 1 mL pipette to mix the solution about 8 to 10 times to ensure complete dissolution of the XAV939 powder. We aliquoted about 20 μL of the 1 mg/mL solution into sterile 1.5 mL sterile microcentrifuge tubes and stored them for up to six months at −20 °C to prevent freeze–thaw. We added 15.6 μL of XAV939 (1 μM) to 50 mL 20% FBS differentiation medium for differentiation. This medium should be freshly prepared and protected from light. This medium was filtered through the 50 mL Steriflip filtration system for additional sterility.

**Gelatin**: The gelatin solution was kept briefly at room temperature to thaw. Under sterile conditions, we prepared 100 mL 0.1% gelatin by diluting 5 mL of sterile stock gelatin with 95 mL of 1× PBS. For an added precaution, we filter-sterilized the solution with a 200 mL stericup filtration system into a sterile bottle. The solution was stored at 4 °C.

**Formaldehyde, 4% (*v*/*v*)**: To prepare a 5 mL solution, we added 541 μL of 37% formaldehyde into 4.46 mL of 1× PBS in the dark. This solution should be freshly prepared.

**PBS (Phosphate Buffered Saline)**: Under sterile conditions, we prepared 1× PBS by mixing 50 mL 10× PBS with 450 mL distilled water. The solution was filtered through a 500 mL stericup filtration system into a sterile 500 mL bottle. PBS should be stored at room temperature.

**Blocking solution (5% Nonfat Dry Milk, 0.4% Triton X-100)**: We added 2.5 g of nonfat dry milk and 400 μL of 50% (*v*/*v*) Triton X-100 solution into 50 mL of 1× PBS. This solution should be freshly prepared.

### 2.4. Procedure

All the procedures described here were undertaken in a sterile biosafety II tissue culture hood. We strictly followed aseptic techniques to prevent contamination. The UV lamp was used for at least 30 min before using the tissue culture hood, and the benchtop was swabbed with 70% ethanol. Anything entering the tissue culture hood was swabbed with 70% ethanol. All pipettes and materials used in the biosafety cabinet were sterile.

### 2.5. Thawing Feeder-Free ESCs

A total of 2 mL of 0.1% gelatin was added to coat the surface of T25 tissue culture flasks. The excess gelatin solution was aspirated off before seeding ESCs.We added 9.5 mL of 37 °C ES + LIF medium into a 15 mL sterile conical tube.A frozen vial of mESCs was removed from liquid nitrogen and the vial was thawed in a 37 °C water bath. The vial was swirled gently until only a small ice crystal remained in the core of the vial.The vial was swabbed with 70% ethanol and placed in the tissue culture hood. Using a sterile 1 mL pipette, we gently transferred the cells into the 15 mL conical tube containing ES + LIF medium.The cells were pelleted by centrifugation for 3 min at 1200 rpm in a benchtop centrifuge at room temperature. The supernatant was aspirated and discarded with a sterilized glass Pasteur pipette.The cell pellet was gently resuspended in 10 mL pre-warmed ES + LIF medium and we transferred the cell suspension into a 25 cm^2^ tissue culture flask. The cells were grown at 37 °C in a humidified 5% CO_2_ incubator.The cells were distributed evenly in the flask by rocking the flask in a gentle left, right, up, and down motions.The next day, the medium was changed with 10 mL ES + LIF medium to remove dead cells and residual DMSO (footnote: pre-warmed medium was added perpendicular to the plate to prevent dislodging cells from the tissue culture plate).

### 2.6. Passaging ESCs Using Trypsin

When ESCs reached about 80–90% confluency (usually about 2–3 days after thawing), we aspirated the old ES + LIF medium and washed with 10 mL room temperature 1× PBS. The PBS was aspirated and the wash was repeated one more time (footnote: the aspirator pipette was positioned away from the cells at the corner of the flask to prevent dislodging the cells).The cells were covered with 1 mL of trypsin solution and incubated at 37 °C for 1–2 min for the cells to be uniformly dispersed into small clumps and detached from the plate surface.We added 9 mL of pre-warmed ES + LIF medium to inactivate the trypsin and gently mixed it about 10 times by pipetting up and down using a 10 mL sterile serological pipette.The cells were counted by mixing 10 μL cell suspension with 10 μL trypan blue solution at a 1:1 ratio. In total, 10 μL of the mixture was transferred into the cell counter slide and cells counted using the automated cell counter.In total, 3.0 × 10^6^ ESCs were transferred into a gelatin-coated 10 cm tissue culture dish in 10 mL ES + LIF medium.After plating the cells, the plate was returned to the incubator. The plate was moved in quick, short, left, right, up, and down motions to evenly disperse the cells across the surface of the plate.The medium was changed the next day by aspirating it and replacing it with a pre-warmed ES + LIF medium.The cells were passaged again after the cells reached about 80–90% confluency by repeating steps 8–14 before proceeding with cardiomyocyte differentiation.

### 2.7. Differentiation of ESCs into Cardiomyocytes

After the cells have reached 80–90% confluency, they can be differentiated into cardiomyocytes. Before performing the steps below, the 20% FBS differentiation medium was prewarmed at 37 °C and supplemented with CHIR990221. This medium was protected from light.Three sterile 10 cm petri plates were filled with about 15 mL of 1× PBS each to flood the base.Steps 8–9 were repeated to detach ESCs from the tissue culture plate. The detached cells were resuspended in 10 mL of 20% FBS differentiation medium supplemented with CHIR99021 by gently pipetting up and down 10 times.Cells were counted by repeating step 11 and diluted in the differentiation medium such that a 20 µL droplet contained 500 cells.Using a 100 µL multichannel pipette with 6 sterile pipette tips attached, we dispensed 20 µL droplets into the lid of the untreated petri plate. The droplets were evenly spaced so they did not touch each other. Each 10 cm petri plate lid contained approximately 60 droplets and were about 1 cm from the edge.The lid was carefully inverted over the base of the petri plate containing PBS. A quick flip-over was done to prevent the drops from running into each other.The plates were moved gently into the cell culture incubator for 48 h. While moving plates, we ensured the PBS solution at the base did not dislodge droplets from the lid.Exactly after 48 h, the plates were returned to the tissue culture hood. We pre-warmed the 20% FBS differentiation medium supplemented with XAV939.Using a 1000 µL pipette, the embryoid bodies were collected into a sterile 15 mL conical tube. These embryoid bodies are very delicate and were gently pipetted to maintain their structural integrity. The embryoid bodies were allowed to settle to the bottom of the conical tube by gravity.The supernatant was gently aspirated and discarded without disturbing the settled embryoid bodies.A total of 2 mL of pre-warmed 20% FBS differentiation medium supplemented with XAV939 was dispensed into 6 cm sterile petri plates and 1 mL of the medium was gently added to the embryoid bodies.Using a 1000 µL sterile pipette, embryoid bodies were gently resuspended and transferred into the 6 cm plate containing the pre-warmed medium.The plate was gently moved into the cell culture incubator for 48 h, and the plate moved in short left, right, up, and down motions to evenly disperse the embryoid bodies across the surface of the plate.After 48 h, the well of the 6-well tissue culture plate was flooded with 2 mL of 0.1% gelatin and placed in the cell culture incubator for about 30 min. Following incubation, the excess gelatin solution was aspirated and discarded.Using a 1000 µL pipette, embryoid bodies were transferred into a 15 mL conical tube and allowed to settle by gravity. After settling, the supernatant was aspirated and discarded without dislodging the embryoid bodies.The embryoid bodies were gently resuspended in 2 mL 20% FBS differentiation medium and transferred into the gelatin-coated well in the 6-well tissue culture plate.The plates were returned to the cell culture incubator at 37 °C for 24 h. *The embryoid bodies will attach and undergo morphological changes*.After 24 h, the 20% FBS differentiation medium was replaced with 3 mL low FBS differentiation medium. This will reduce the growth of other cell types.The medium was changed with fresh pre-warmed low FBS differentiation medium every 24 h and cells were monitored for contractile activity.Typically, contractile activity was observed after 8 days post-differentiation.

### 2.8. Characterization of Cardiomyocytes and Differentiation

#### 2.8.1. RT-QPCR for Stage-Specific Gene Markers during Cardiomyocyte Differentiation

In total, 2–5 × 10^6^ cells were used for RNA purification from ESCs before differentiation. For samples from days 2, 4, 6, 8, and 12 post-differentiation, we prepared four 10 cm petri plates each for RNA purification. The procedure described here uses TRizol for RNA purification on a clean lab bench. RNA is very unstable and prone to degradation by RNases that are ubiquitously present in the lab environment. Therefore, we used autoclaved pipette tips, a clean bench, and RNaseAway to clean pipettes before RNA purification. The microcentrifuge tubes were handled with clean gloves, and touching the inner lid of the tubes was avoided as this could introduce RNases to the samples.

For undifferentiated cells, 5 × 10^6^ cells were pelleted by centrifugation at 1200 rpm for 3 min, supernatant discarded, cells washed with 1× PBS, and 1 mL of TRIzol^®^ reagent added. For embryoid bodies on day 2 and day 4, embryoid bodies were pelleted by gravity, washed with 1× PBS, and 1 mL of TRIzol^®^ reagent added. For differentiation days 6, 8, and 12, cells were detached using a cell lifter and pelleted by centrifugation at 1200 rpm for 3 min or by washing attached cells gently with PBS and directly adding 1 mL of TRIzol^®^ reagent.Using a 1 mL pipette, the TRIzol^®^ reagent was mixed with the sample until there were no visible cells by pipetting up and down about 10 times. The samples were then incubated for 5 min at room temperature.A total of 200 µL of chloroform was added to the tubes and shaken vigorously by hand for 15 s and allowed to settle for 3 min at room temperature.After incubation, the phases began to separate. The tube was gently placed in a centrifuge and spun at 12,000× *g* for 15 min at 4 °C.At this point, the phases were well separated, and care was taken when removing tubes from the centrifuge to prevent mixing the two phases. The top aqueous phase (~500 µL) was collected into another clean 1.5 mL tube.A total of 500 µL of 100% isopropanol was added to the aqueous phase, mixed a few times by inverting the tubes up and down, and incubated at room temperature for 10 min. The tubes were centrifuged at 12,000× *g* for 10 min at 4 °C.The supernatant was discarded from the tube and the RNA pellet washed with 1 mL of 75% ethanol. The tube was centrifuged at 7500× *g* for 5 min at 4 °C, the wash discarded, and the RNA pellet air dried for 10 min.The RNA pellet was resuspended in 85 µL RNAse-free water, 4 µL DNAse, 1 µL RNase inhibitor, and 10 µL of 10× DNase Buffer was added. The resuspended RNA was mixed properly and incubated at 37 °C for 2 h.Following DNase treatment, the RNA was cleaned up by either (A) repeating steps (1–7) or following the instructor manual of the PureLink RNA Mini Kit.After final purification and resuspension of RNA in ~100 µL of RNase-free water, 0.5 µL of RNase inhibitor was added.The integrity of RNA was assessed by running 1 µg of RNA on an agarose gel. The RNA was quantified before proceeding to the next steps.cDNA was synthesized using 1 µg of RNA and random hexamers by following the instructor’s manual of the RT-cDNA synthesis kit.For quantitative PCR, 1:50 cDNA dilution was prepared. In total, 6 µL of the cDNA dilution, 1.5 µL of primer mix (forward and reverse), and 7.5 µL of Evagreen^®^ mix was used per well. The qPCR reaction was performed in the quantitative-PCR detection system following the Evagreen^®^ instructor’s manual.

#### 2.8.2. cTnT Immunostaining Analysis for Cardiomyocytes

Day 12 differentiated cardiomyocytes were washed with 1 mL PBS in the 6-well tissue culture plates.In the dark, 1 mL of freshly prepared 4% (*v*/*v*) formaldehyde was added and incubated for 20 min at room temperature to fix cells.Following incubation, we washed the cells 3 times with 1× PBS, aspirating PBS between each wash.We prepared the primary antibody solution by adding 1 µL of cTnT antibody to 1 mL of blocking solution, mixed it properly, and added it to the well containing fixed cardiomyocytes.The cells were incubated with cTnT antibody overnight at 4 °C or room temperature for 2 h.Following incubation, the cTnT antibody solution was discarded and the cells washed with 1× PBS three times, aspirating PBS between each wash.In the dark, the secondary antibody solution was prepared by diluting 1 µL of mouse IgG Alexa Fluor^®^ 488 in 1 mL blocking solution. This solution was added to the well containing fixed cells and incubated for 30 min at room temperature in the dark.The secondary antibody solution was aspirated in the dark and step 6 repeated.Following washes, 1 mL of 0.5 µg/mL DAPI solution in PBS was added to the well in the dark.Immunofluorescence images were taken using the EVOS^®^FL fluorescence microscope.

Footnote: The EVOS^®^FL microscope can take images of plates directly. Alternatively, cardiomyocytes can be grown on coverslips in the wells and transferred onto a microscope slide after following steps 1–9 described above. The cover slip should be mounted over the mounting medium on a microscope slide.

### 2.9. Fluorescence-Assisted Cell Sorting (FACS) Analysis of Cardiomyocytes

#### 2.9.1. Preparation of Cells for FACS

We performed quantitative analysis to determine the purity and yield of cardiomyocytes generated using the combination of CHIR99021 and XAV939 by following previously published methods [26]. Briefly, cardiomyocytes on day 12 post-differentiation were washed with 1× PBS to remove the medium and dead cells. The cardiomyocytes were then incubated with 1 mL 0.25% Trypsin-EDTA for 5 min at 37 °C to singularize cells. The trypsin reaction was quenched with the differentiation medium containing FBS and cells further singularized by continuous pipetting. The cells were counted and pelleted at 200× *g* for 5 min. The cell pellet was resuspended in 1% formaldehyde for 20 min at room temperature. The cells were pelleted and resuspended in cold 90% methanol for 15 min at 4 °C. The cells were pelleted again, and 0.5 million cells were resuspended in 1× PBS buffer containing 2.5% BSA to wash cells. This wash was performed 3 times to remove residual methanol. Next, the cells were resuspended in 1× PBS buffer containing 2.5% BSA and 0.1% Triton X-100 without antibody or with cTnT antibody at a dilution of 1:100 and incubated overnight at 4 °C. Following incubation, the cells were washed three times with 1× PBS buffer containing 2.5% BSA and 0.1% Triton X-100. Here, cells were incubated with no secondary antibody or AlexaFluor 488 secondary antibody at a 1:1000 dilution for 30 min at room temperature. The cells were washed three times with 1× PBS buffer containing 2.5% BSA and 0.1% Triton X-100. We then performed FACS using the BD Fortessa.

#### 2.9.2. FACS Analysis

We collected data for unstained, only secondary antibody staining, and both primary antibody (cTnT) and secondary antibody staining. We used cells collected from secondary antibodies only as a background for gating. These were referred to as cTnT negative cells. The cells collected outside this gate following both cTnT and AlexaFluor 488 incubations were gated as cTnT-positive cells.

### 2.10. Summary of Cardiomyocyte Differentiation Using WNT Switch Method

Briefly, mouse ESCs were cultured for at least two passages after thawing using the medium containing the leukemia inhibitory factor (LIF). To begin differentiation, the cells were cultured in 20% differentiation medium supplemented with CHIR99021 as hanging drops. Uniform, rounded embryoid bodies were formed on day 2 (D2) of differentiation (Figure 1). The embryoid bodies were transferred to petri plates in the differentiation medium supplemented with XAV939. On day 4 of differentiation (D4), an increase in embryoid body size was observed, with cells beginning to attach to the surface of the petri plates (Figure 1). After transferring embryoid bodies to gelatinized plates, significant morphology changes were observed, with cells growing out of the embryoid body core to generate a monolayer sheet of attached cells by day 8 (D8). These sheets of cells also displayed contractile activity, which continued to increase through day 12 (D12) of differentiation (Figure 1, Appendix A).

## 3. Results

### 3.1. Identification of Optimum CHIR99021 Concentration for WNT Switch Method

CHIR99021 has previously been used for cardiomyocyte differentiation from human IPSCs. Therefore, to identify the optimum CHIR99021 concentration sufficient to induce cardiomyocyte differentiation from mouse ESCs, we used 1, 2, 3, and 4 μM CHIR99021 for the first two days of differentiation. The treatment with CHIR99021 was immediately followed by XAV939 treatment. In addition, we also differentiated mouse ESCs into cardiomyocytes using the previously published method, which only utilized the XAV939 treatment [30]. We then characterized the cardiomyocytes generated under these conditions by monitoring the expression of the cardiomyocyte-specific genes (*cTnT*, *Myh6*, and *Nkx2.5*) on days 8 and 12 post-differentiation by RT-qPCR. We observed the least induction of *cTnT*, *Myh6,* and *Nkx2.5* from cells differentiated with 1 µM CHIR99021. In both day 8 and day 12 post-differentiation, treatment with 4 µM CHIR99021 showed the highest *Nkx2.5* gene expression. On day 8 of cardiomyocyte differentiation, the expression of *cTnT* in 2, 3, and 4 µM CHIR99021 treated cells was similar to cells only treated with XAV939 (Figure 2A). Interestingly, the expression of *cTnT* and *Myh6* was highest in 3 µM treated cells compared to that in other conditions (Figure 2A,B). More importantly, the induction of these cardiomyocyte-specific gene markers was significantly higher in the 3 μM CHIR99021 treated samples compared to the only XAV939 treated cells (Figure 2A–C). These data suggest that 3 μM CHIR99021 treatment of ESCs for two days post-differentiation followed by treatment with XAV939 is sufficient to induce higher yields of cardiomyocytes.

### 3.2. WNT Switch Method Causes Pluripotency Genes’ Repression during Cardiomyocyte Differentiation

The successful differentiation of pluripotent stem cells requires the complete repression of pluripotency-specific genes post-differentiation. Therefore, we asked whether the expression of the pluripotency master regulators *Oct3/4* and *Nanog* were repressed post-differentiation. We purified total RNA from undifferentiated and days 2, 4, 6, 8, and 12 cells post-differentiation. Following cDNA synthesis, we performed quantitative qPCR using gene-specific primers for *Oct3*/*4* and *Nanog*. We observed a significant repression of *Oct3/4* expression beginning from day 6 post-differentiation. The expression of *Oct3/4* continued to decrease significantly by day 8 and day 12 post-differentiation (Figure 3A). Additionally, the expression of *Nanog* was immediately repressed by day 2 post-differentiation, further decreasing the expression throughout the cardiomyocyte differentiation (Figure 3B). Taken together, the expression profiles of the pluripotency-specific genes suggest a complete exit from the pluripotency state post-differentiation.

### 3.3. WNT Signaling Gene Expression Responds to CHIR99021 and XAV939 Treatments

Our method relies on modulating the WNT signaling pathway to induce mesoderm and cardiac mesoderm formation. We expected to achieve increased WNT signaling upon the treatment of cells with CHIR99021 and the inhibition of the WNT signaling pathway by treatment with XAV939. To determine whether we successfully induced WNT signaling and its inhibition thereof, we measured the expression of WNT signaling genes at days 2 and 4 post-differentiation compared to undifferentiated cells. The genes we chose to monitor for the WNT signaling pathway also indicate mesoderm differentiation. As expected, we saw a significant increase in the expression of *Gata4* (Figure 4A), *Lhx1* (Figure 4B), *Wnt3* (Figure 4C), and *Brachyury* (Figure 4D), indicating an induction of the WNT signaling pathway by CHIR99021. Similarly, the treatment of cells with XAV939 reduced the expression of these genes by day 4 post-differentiation (Figure 4A–D). Taken together, these data highlight the efficiency of our method in inducing WNT signaling and mesoderm genes, which is critical for cardiomyocyte differentiation.

### 3.4. The WNT Switch Method Increases Cardiomyocyte Yields

A key feature of cardiomyocytes is their unique morphology and sarcomere structures. This morphology can be visualized using immunostaining against cTnT, cTnI, MLC2a, and α-actinin. To characterize the efficient differentiation of ESCs into cardiomyocytes using our method, we performed immunostaining against cTnT on day 12 post-differentiation. Our data showed a positive immunostaining of cTnT in day 12 cardiomyocytes (Figure 5A). Additionally, we measured the contractile activity of the cardiomyocytes by measuring the beats per minute (bpm) using videos obtained from brightfield microscopy. We observed a mean bpm of about 40 and 60 at D8 and D12 respectively, which are similar to rates previously reported [31].

Next, we compared the yield of cardiomyocytes generated by the WNT Switch method with a currently used method [30], which utilizes only XAV939 treatment using fluorescence-activated cell sorting (FACS). Briefly, cardiomyocytes were dissociated, fixed, and stained with anti-cTNT and AlexaFluor-488 antibodies. As a negative control, we stained dissociated cardiomyocytes with the AlexaFluor-488 antibody without the anti-cTnT antibody (Figure 6A,C). Following FACS analysis, the induction of cardiomyocytes from XAV939 treated cells yielded about 57.1% cardiomyocytes (Figure 6B), similar to the 58.7% yield previously published [30]. As expected, our method for cardiomyocyte induction generated about 86.1% yield (Figure 6D). The 30% increase in the yield of cardiomyocytes demonstrates the robustness of the WNT Switch method.

## 4. Discussion

The heart is the first organ formed during mammalian development and is controlled by gene regulation and signal transduction pathways. In particular, the WNT signaling pathway is critical for the efficient development of cardiomyocytes [27]. These cardiomyocytes confer function to the heart through their contractile activity and are a critical component of the heart. In fact, due to their reduced proliferative ability, cardiomyocyte deficiency underlies most heart failures [3,4,5]. Here, we describe the WNT Switch method as an efficient way to generate cardiomyocytes from mouse ESCs in vitro that could be used to study the molecular mechanisms regulating cardiomyocyte development. This method leverages the regulation of WNT signaling during normal mammalian heart development. Cardiomyocyte progenitors originate from the mesoderm germ layer following gastrulation [32]. Thus, it is imperative to induce mesoderm formation during cardiomyocyte differentiation. Moreover, strong evidence supports the role of the WNT signaling pathway in inducing mesoderm formation [33,34,35]. Previous methods developed for mouse cardiomyocyte differentiation rely on the endogenous activation of WNT signaling or use of growth factors to induce mesoderm post-differentiation [6,30,36,37]. The WNT Switch method is distinct, given that we treat cells with CHIR99021 during embryoid body formation to induce mesoderm formation. Upon treatment of ESCs from undifferentiated to day 2 post-differentiation, we show an increased expression of mesoderm-specific genes (Figure 4), suggesting an induction of mesoderm formation. The high expression of the WNT3 ligand (*Wnt3a*) and the WNT response genes (*T*, *Lhx1*, *Gata4*) supports the specific induction of the WNT signaling pathway. The relative expression of the mesoderm-specific genes we obtained from our differentiation method is comparable to other methods that utilize expensive but unstable growth factors [7].

A properly formed mesoderm is a hub for cardiomyocyte progenitor cells and generates progenitor cells for endothelial and hematopoietic lineages [38,39]. Interestingly, the WNT signaling pathway contributes to the fate of mesoderm cells into distinct lineages [40]. The continuous activation of WNT signaling is implicated in the formation of endothelial cells and other mesodermal lineages [41,42,43]. However, inhibiting the WNT signaling pathway promotes cardiomyocyte progenitor lineages post-mesoderm differentiation [16,17,24,26]. Therefore, we immediately followed the activation of mesoderm formation with the inhibition of WNT signaling using XAV939 from day 2 to day 4 post-differentiation. We observed a decrease in the gene expression of the WNT signaling genes on day 4 (Figure 4A–D).

The benchmark for cardiomyocyte differentiation is spontaneous and uniform contractile activity. We began observing spontaneous contractile activity from day 8 post-differentiation, which continued to increase by day 12 of differentiation (Figure 5B, Appendix A). The presence of contractile activity corroborated the elevated expression of the cardiac-specific gene markers, which is higher than in previously reported methods (Figure 2) [7,30]. We also obtained cardiomyocyte-specific positive cTnT immunostaining, confirming successful differentiation (Figure 5A). The increased expression of the cardiac-specific gene markers and the uniform contractile activity highlights the success of our method. Additionally, we show that the combined treatment with CHIR99021 and XAV939 increased the yield of cardiomyocytes (86.1%) compared to previous methods, including the XAV939 treatment alone (57.1%) (Figure 6) [6,36,37]. While we used gene expression analysis and contractile activity to characterize cardiomyocytes, the functional activity is usually measured by functional assays such as calcium imaging. The intracellular calcium imaging method utilizes calcium-sensitive fluorescent dyes like fura-2 and confocal microscopy to measure the calcium sparks underlying cardiomyocyte contractile activity [44,45,46]. Other methods include patch clamp and traction force microscopy [47].

## 5. Conclusions

Altogether, mESCs are auseful differentiation model that has propelled discovery of early developmental processes and their role in various diseases. The WNT Switch method uses mESCs to generate cardiomyocytes in a fast, efficient, and reproducible way This method will advance studies on understanding molecular mechanisms and gene regulatory factors that control the cardiomyocyte development. For example, molecular experiments like chromatin immunoprecipitation (ChIP) at the different stages during cardiomyocyte differentiation often require a large number of cells, and the WNT Switch method can generate sizable cardiomyocytes for molecular analysis. Additionally, the generation of a homogeneous population of cardiomyocytes can help identify therapeutic targets that could unlock the proliferative ability of cardiomyocytes in vivo.

## Figures and Tables

**Figure 1 cells-13-00132-f001:**
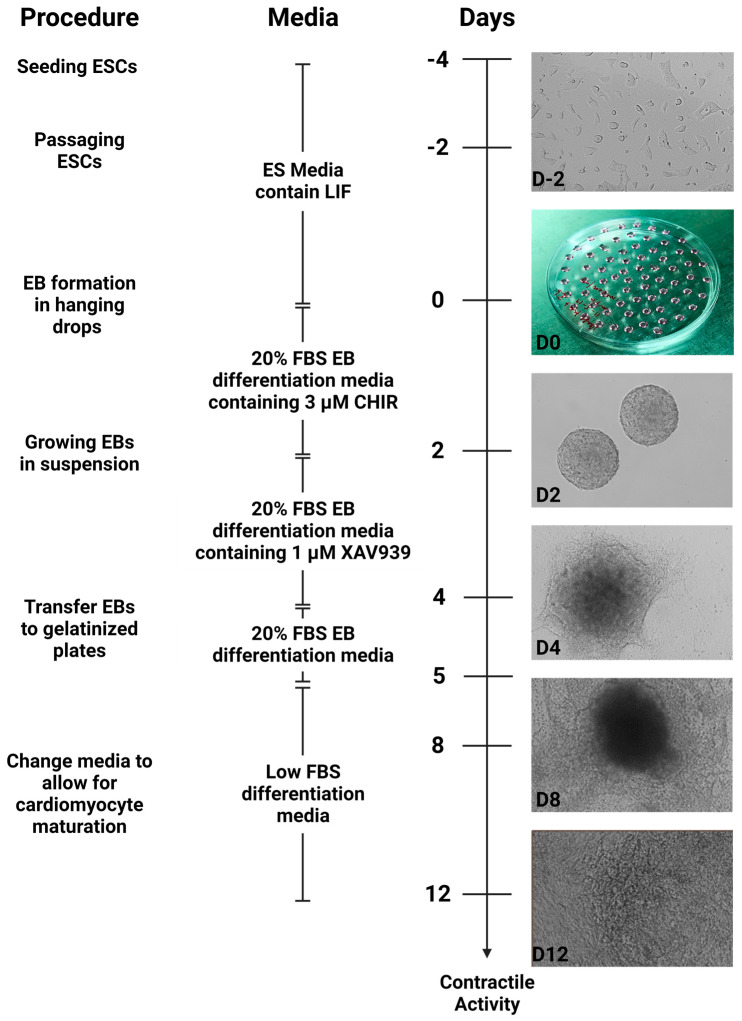
Schematic of WNT Switch method for differentiating mouse ESCs into cardiomyocytes: the schematic summarizes the differentiation method for ESCs into cardiomyocytes. D2 to D12 are the days of differentiation. Representative images of the differentiation states are shown along withthe timeline at 10× magnification.

**Figure 2 cells-13-00132-f002:**
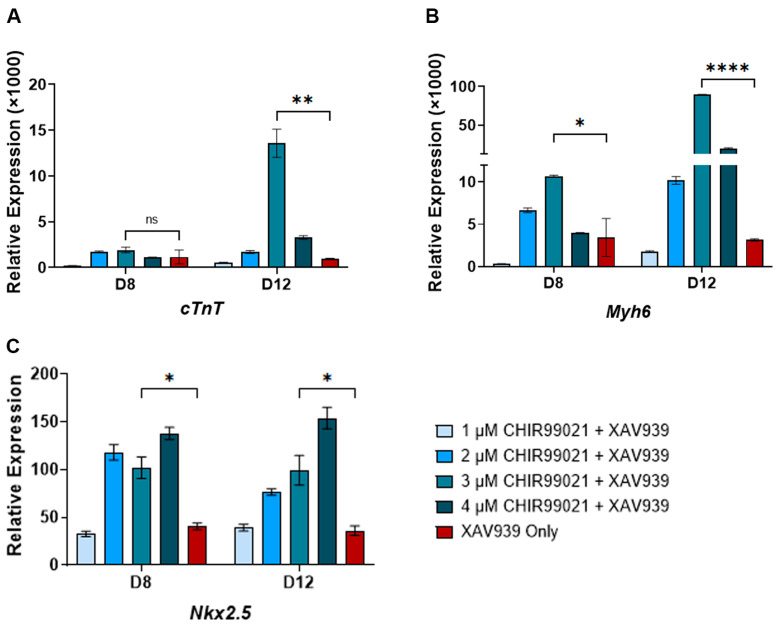
RT-qPCR for cardiac-specific genes in undifferentiated and differentiated cells.: (**A**) expression of *cTnT* gene; (**B**) expression of the *Myh6* gene; (**C**) expression of *Nkx2.5.* The data show the gene expression for cardiac-specific genes relative to *Gapdh* expression. The relative expression was normalized to gene expression in the undifferentiated cells, which has been set to 1. The mathematical equation used was 2^−ΔΔCq^, where ΔΔcq represents (*Gapdh*—gene target) and (undifferentiated—day of differentiation). UD: undifferentiated, D2–D12: days post-differentiation, Cq: quantification cycle obtained from the qPCR machine. Error bars represent the standard error of the mean from n ≥ 3 independent experiments. ns: non-significant change, ****: *p*-value < 0.0001, **: *p*-value < 0.01, *: *p*-value < 0.05. Statistical significance was obtained by performing a student *t*-test comparing gene expression in 3 µM CHIR99021 treated cells to XAV939 only cells.

**Figure 3 cells-13-00132-f003:**
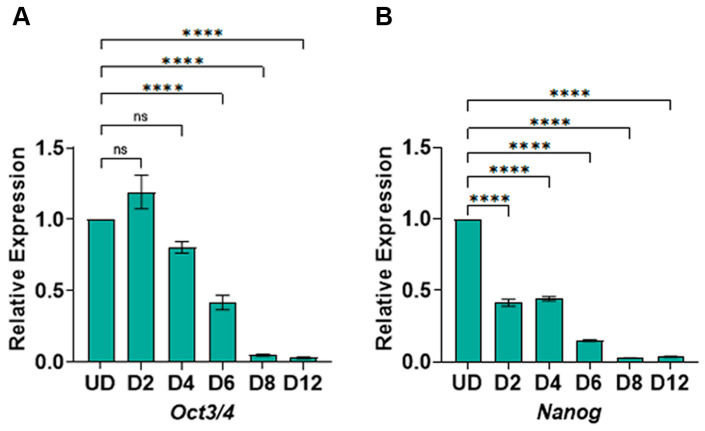
RT-qPCR for pluripotency genes in undifferentiated and differentiated cells: (**A**) expression of *Oct3/4* gene; (**B**) expression of the *Nanog* gene. The data show the gene expression for *Oct3/4* and *Nanog* relative to *Gapdh* expression. The relative expression was then normalized to gene expression in the undifferentiated cells, set to 1. The mathematical equation used was 2^−ΔΔCq^, where ΔΔcq represents (*Gapdh*—gene target) and (undifferentiated—day of differentiation). UD: undifferentiated, D2–D12: days post-differentiation, Cq: quantification cycle obtained from the qPCR machine. Error bars represent the standard error of the mean from n ≥ 3 independent experiments. ns: non-significant change, ****: *p*-value < 0.0001. The statistical significance was obtained by performing unpaired *t*-test in GraphPad prism comparing gene expression in days post-differentiation to the undifferentiated cells.

**Figure 4 cells-13-00132-f004:**
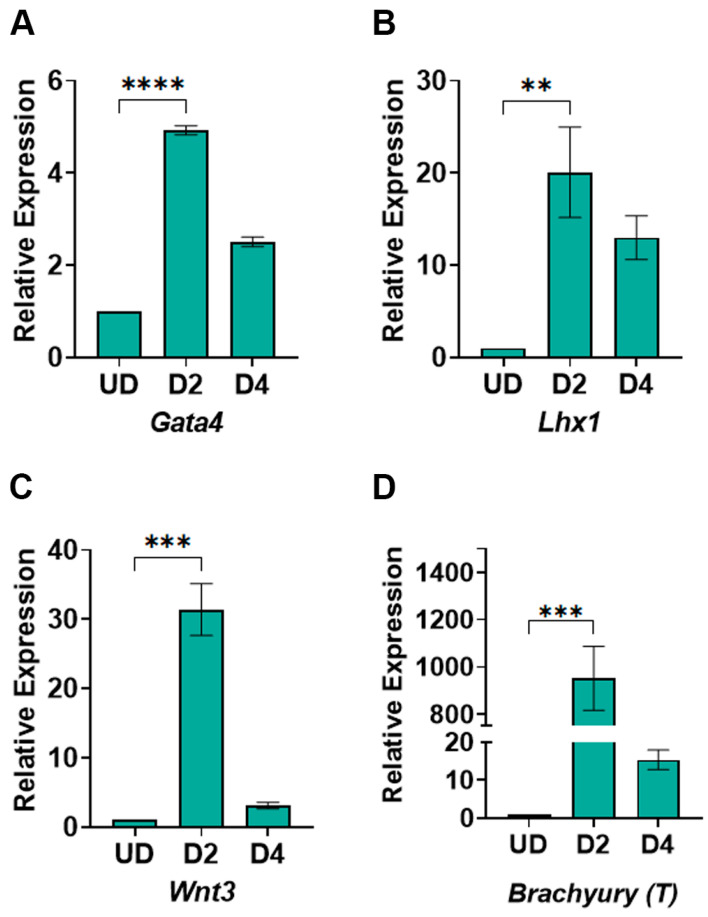
RT-qPCR for mesoderm-specific genes in undifferentiated and differentiated cells: (**A**) expression of the *Gata4* gene; (**B**) expression of the *Lhx1* gene; (**C**) expression of *Wnt3*; (**D**) expression of *Brachyury (T).* The data show the gene expression for mesoderm-specific genes relative to *Gapdh* expression. The relative expression was normalized to gene expression in the undifferentiated cells, which has been set to 1. The mathematical equation used was 2^−ΔΔCq^. where ΔΔcq represents (*Gapdh*—Gene target) and (Undifferentiated—day of differentiation). UD: Undifferentiated, D2–D4: days post-differentiation, Cq: quantification cycle obtained from the qPCR machine. Error bars represent the standard error of the mean from n ≥ 3 independent experiments ****: *p*-value < 0.0001, ***: *p*-value < 0.001, **, *p*-value < 0.01. Statistical significance was obtained by performing unpaired *t*-test in GraphPad prism comparing gene expression in day 2 post-differentiation to the undifferentiated cells.

**Figure 5 cells-13-00132-f005:**
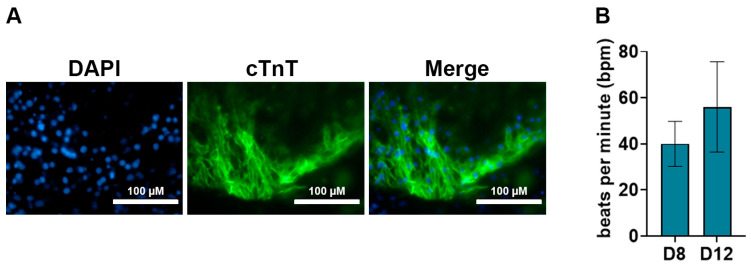
Characterization of cardiomyocytes. (**A**) Immunostaining of day 12 cardiomyocytes showing a positive immuno-stain against anti-cTnT antibody. DAPI was used for nucleus staining. The scale bar represents 100 μM. (**B**) Average beating rates of cardiomyocytes at days 8 and 12 post-differentiation. Bpm: beats per minute.

**Figure 6 cells-13-00132-f006:**
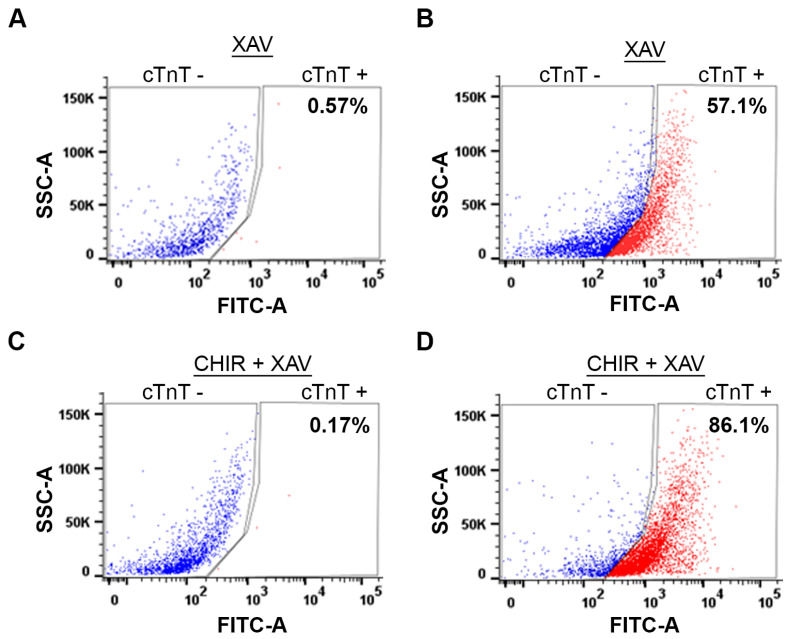
Induction of cardiomyocytes by WNT Switch method yields increased cardiomyocytes: (**A**,**B**) representative FACS analysis of cardiomyocytes generated from only XAV939 treatment in (**A**) absence of cTnT antibody and (**B**) with cTnT antibody. Representative FACS analysis of cardiomyocytes generated from 3 µM CHIR99021 and XAV939 treated cells in (**C**) the absence of cTnT antibody and (**D**) in the presence of cTnT antibody. XAV: XAV939, CHIR: CHIR99021, Blue points: cTnT negative cells, red points: cTnT positive cells.

## Data Availability

The original contributions presented in the study are included in the article/Appendix A, further inquiries can be directed to the corresponding author.

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
