# Peer review of "Cardiomyocyte Differentiation from Mouse Embryonic Stem Cells by WNT Switch Method"

_cells, 2024, doi:10.3390/cells13020132_

Round 1

Reviewer 1 Report (Previous Reviewer 1)

Comments and Suggestions for Authors

Thank you so much.

All the comments are sufficiently explained

Author Response

We thank the reviewer for accepting our changes and for helping us improve the manuscript.

Reviewer 2 Report (Previous Reviewer 2)

Comments and Suggestions for Authors

Dear Editors,

The authors have adequately addressed the questions raised. By comparing the present work to the literature and by assessing the cardiomyocyte differentiation percentage, the importance of the described technique increased significantly.

It would be preferable, though, to include a brief perspective note within the discussion, on the use of functional experiments (i.e. calcium imaging) to better characterize the phenotype of the cells.

Kind Regards

Author Response

Thank you for your helpful comments. W have added the suggested information in the discussion section, which is highlighted.

This manuscript is a resubmission of an earlier submission. The following is a list of the peer review reports and author responses from that submission.

Round 1

Reviewer 1 Report

Comments and Suggestions for Authors

The authors did great work developing a method for producing CM from mESC. They claim that this method is affordable and results in good-quality CM. but 

1) the authors did not explain the novelty of their method compared to other published methods.

2) the authors did not justify the quantity of drugs that they decided to use. For example, they should perform a dose-dependent curve and decide the best dose

3) The number of experiments is not stated to ensure the reproducibility of the method

4) does the error bar indicate S.D or S.E

6) they need more evidence to characterize the newly synthesized CMs

7) they claim that this produces robust CM differentiation?? How did they claim that? What is the percentage of the CM produced?

8) they need to discuss the role of Wnt signaling during the differentiation.

Comments on the Quality of English Language

the paper is very well written but the style of writing needs to be improved to fit the quality of a method paper

Reviewer 2 Report

Comments and Suggestions for Authors

Dear Editors,

The following study submitted to Cells describes a protocol for the differentiation of mouse embryonic stem cells into cardiomyocytes using small-molecule WNT modulators, emphasizing its significance as a model system for studying cardiac development, modeling diseases, and drug screening. The protocol is extensively detailed and well-structured.

There are some concerns that arise and need to be addressed.

Major issues

-        CHIR99021 is a WNT activator not an inhibitor. Thus, title, abstract, graphical abstract, and the rest of the manuscript should be rectified accordingly.

-        In the introduction section, the authors mention thatPrevious methods for differentiating mouse ESCs (mESCs) into cardiomyocytes use growth factors, which are expensive and have short half-lives”; however, they use a similar technique by forming embryoid bodies via the hanging-drop method. Besides, there is a plethora of studies that used embryoid bodies to obtain cardiomyocytes. Based on this, a clear comparison between the present technique and literature is required.

-        CHIR99021 has been extensively used on human pluripotent stem cells and human induced pluripotent stem cells to generate cardiomyocytes; please add and discuss representative literature on this matter in the manuscript.

-        What is the beating percentage of the cells? In other terms, what is the differentiation percentage efficiency of this method?

-        It would be preferable if the authors show some calcium transients, if possible, in these generated cardiomyocytes.

Minor issues

-        In line 12, please rectify CHIR99021.

-        Are CHIR99021 and XAV939 effects dose dependent?

-        Line 82: rectify “fluorescence”

-        What is the rationale of using the starvation media? Kindly make note of this in the methods.